# The Cellular Innate Immune Response of the Invasive Pest Insect *Drosophila suzukii* against *Pseudomonas entomophila* Involves the Release of Extracellular Traps

**DOI:** 10.3390/cells10123320

**Published:** 2021-11-26

**Authors:** Tessa Carrau, Susanne Thümecke, Liliana M. R. Silva, David Perez-Bravo, Ulrich Gärtner, Anja Taubert, Carlos Hermosilla, Andreas Vilcinskas, Kwang-Zin Lee

**Affiliations:** 1Department Pests and Vector Insect Control, Fraunhofer Institute for Molecular Biology and Applied Ecology, Ohlebergsweg 12, D-35394 Giessen, Germany; tessa.carrau@gmail.com (T.C.); andreas.vilcinskas@ime.fraunhofer.de (A.V.); 2Institute for Insect Biotechnology, Justus Liebig University, Heinrich Buff Ring 26-32, D-35392 Giessen, Germany; susanne.thuemecke@uni-rostock.de; 3Institute of Parasitology, Justus Liebig University, Schubert Strasse 81, D-35392 Giessen, Germany; anja.taubert@vetmed.uni-giessen.de (A.T.); carlos.r.hermosilla@vetmed.uni-giessen.de (C.H.); 4Department of Internal Medicine (Pulmonology), University of Giessen and Marburg Lung Center (UGMLC), Member of the German Center for Lung Research (DZL), Aulweg 123, D-35394 Giessen, Germany; david.perez.bravo89@gmail.com; 5Institute of Anatomy and Cell Biology, Justus Liebig University, Aulweg 123, D-35392 Giessen, Germany; ulrich.gaertner@anatomie.med.uni-giessen.de

**Keywords:** cell culture, *Drosophila suzukii*, hemocytes, plasmatocytes, extracellular traps

## Abstract

*Drosophila suzukii* is a neobiotic invasive pest that causes extensive damage to fruit crops worldwide. The biological control of this species has been unsuccessful thus far, in part because of its robust cellular innate immune system, including the activity of professional phagocytes known as hemocytes and plasmatocytes. The in vitro cultivation of primary hemocytes isolated from *D. suzukii* third-instar larvae is a valuable tool for the investigation of hemocyte-derived effector mechanisms against pathogens such as wasp parasitoid larvae, bacteria, fungi and viruses. Here, we describe the morphological characteristics of *D. suzukii* hemocytes and evaluate early innate immune responses, including extracellular traps released against the entomopathogen *Pseudomonas entomophila* and lipopolysaccharides. We show for the first time that *D. suzukii* plasmatocytes cast extracellular traps to combat *P. entomophila*, along with other cell-mediated reactions, such as phagocytosis and the formation of filopodia.

## 1. Introduction 

*Drosophila suzukii* Matsumura (Diptera: Drosophilidae), also known as the spotted wing *Drosophila*, is a neobiotic invasive pest native to Asia that has spread all over the world and now infests a broad range of fruit crops [1,2,3,4]. Female flies are equipped with a serrated ovipositor that can penetrate intact fruit skins [5]. Eggs are laid inside intact fruits, protecting the developing larvae from topical pesticides [6,7]. The high reproduction rate and rapid life cycle of *D. suzukii* pose a serious economic threat to fruit and wine production. A natural innate resistance to pathogenic stressors appears to reflect the high hemocyte count of infected individuals and efficient hemocyte recruitment to infection sites [8,9,10]. Hemocytes mediate diverse innate defense mechanisms, such as phagocytosis, degranulation, nodulation and encapsulation, as part of the arthropod innate immune system [11,12].

Hematopoiesis in *Drosophila* species produces two hemocyte populations, one originating from the head mesoderm during early embryogenesis and the other arising later from the mesodermal lymph glands [12]. Differentiation of the embryonic hemocytes into lamellocytes, crystal cells and plasmatocytes occurs during the final stage of embryogenesis. The lamellocytes are an independent hemocyte lineage maintained in low numbers, but the population expands significantly in response to parasitoid wasp invasion [13]. These large flat cells encapsulate invading organisms that are too large to be phagocytosed [14]. Crystal cells are involved in the melanization of pathogens and also produce free radicals, such as reactive oxygen species (ROS) [14]. Plasmatocytes are small, spherical cells capable of phagocytosis. They originate in the procephalic mesoderm and migrate to colonize the entire embryo, making up the majority of all hemocytes in vivo [14,15,16,17,18,19,20]. Plasmatocytes act as macrophages by recognizing and eliminating microorganisms and apoptotic cells [19,20,21].

Hemocytes have been studied extensively in the model organism *Drosophila melanogaster* [13]. However, much less is known about these cells in *D. suzukii*. Similarities to mammalian leukocytes, such as neutrophils, suggest a conserved set of functions (and consequences of dysfunction) [20,21,22,23]. In vertebrates, polymorphonuclear neutrophils (PMNs) are the first leukocytes to arrive at an infection site, where they facilitate the removal of pathogens not only by phagocytosis, ROS production and degranulation, but also by NETosis, the release of neutrophil extracellular traps (NETs) [23]. These extracellular webs are composed mainly of DNA decorated with nuclear histones (H1A, H2A/H2B, H3 and H4) and various antimicrobial molecules [24,25,26]. The release of extracellular traps is not limited to PMNs, but also occurs as a highly conserved mechanism in other vertebrate nucleated immune cells (e.g., monocytes, macrophages, eosinophils and mast cells), as well as their invertebrate counterparts [27,28,29,30,31,32]. In insects, for example, extracellular traps are produced by hemocytes in the larvae of the greater wax moth (*Galleria mellonella*) [32].

The systemic and oral infection of *Drosophila* by the entomopathogenic bacteria *Pseudomonas entomophila* has been shown to be a well-suited model system for the analysis of the insects’ humoral and cellular immune response mechanisms [33]. Assuming the bacterial infection would similarly activate defense responses in *D. suzukii* hemolymph, we investigated the ability of different hemocytes to cast extracellular traps following exposure to live *P. entomophila* cells or lipopolysaccharides (LPSs). We found that the coculture of *P. entomophila* with *D. suzukii* plasmatocytes not only triggered the extrusion of extracellular traps, but also resulted in firm bacterial entrapment. Primary cultures of *D. suzukii* plasmatocytes therefore provide a useful in vitro model for the analysis of insect innate immunity, particularly the formation of extracellular traps.

## 2. Materials and Methods

### 2.1. Preparation of Drosophila suzukii Fly Stocks

Flies were maintained at 26 °C and 60% humidity with a 12 h photoperiod. They were reared on a soybean and cornmeal medium comprising 10.8% (*w*/*v*) soybean and cornmeal mix, 0.8% (*w*/*v*) agar, 8% (*w*/*v*) malt, 2.2% (*w*/*v*) molasses, 1% (*w*/*v*) nipagin and 0.625% propionic acid. To avoid contamination, the food was cooked using a MediaClave 10 media sterilizer (WVR International). Before the experiments, the stock was tested for pathogens as previously described [34], including a panel of viruses that commonly infect *D. suzukii* [35].

### 2.2. Hemocyte Collection and Identification

Third-instar larvae (L3) were washed up to 10 times in distilled water to remove debris. Then, larvae were immobilized and dissected as described by Tracy et al. [36]. The latter protocol was followed for hemocyte isolation with slight modifications: hemocytes were isolated directly in Nunc Lab-Tek II chamber slides (Thermo Fisher Scientific, Schwerte, Germany) containing Grace’s insect medium (Thermo Fisher Scientific, Schwerte, Germany) supplemented with 0.1% (*w*/*v*) phenylthiourea (Merck, Darmstadt, Germany) and 10% fetal bovine serum (Merck, Darmstadt, Germany). Up to 100 larvae per well/condition were required to recover 5000 hemocytes. The hemocytes were allowed to attach to the surface of the chamber for at least 30 min and were then washed several times with sterile PBS to prevent cross-contamination [36]. For morphological characterization, isolated hemocytes were fixed in 4% paraformaldehyde (PFA) for 5 min at room temperature and stained with 1% toluidine blue (Merck, Darmstadt, Germany), and at least 100 hemocytes per sample were counted under a Leica DM4 B microscope (Leica Microsystems, Wetzlar, Germany).

### 2.3. Cell Viability Assay

Hemocytes (*n* = 100) were resuspended in imaging medium, which is Grace’s insect medium containing Hoechst (diluted 1:1000) to label DNA and SYTOX Green (diluted 1:2000), to label dead cells. For 3D holotomography, hemocytes in imaging medium were seeded into 35 mm low-rimmed tissue culture µ-dishes (Ibidi^®^, Gewerbehof, Germany) and allowed to settle for 10–15 min. Images were acquired using a 3D Cell Explorer-fluo microscope (Nanolive^®^, Tolochenaz, Switzerland) equipped with 60× magnification (λ = 520 nm, sample exposure 0.2 mW/mm^2^) and a depth of field of 30 µm and an Ibidi^®^ top-stage chamber (Ibidi^®^, Gewerbehof, Germany ) to keep the temperature stable (RT). At the end of the experiment, images were analyzed using Steve^®^ software v.2.6 (Nanolive^®^, Tolochenaz, Switzerland) to obtain refractive index (RI)-based z-stacks [37]. Further, 3D rendering and digital staining were performed based on RI values and thereafter illustrated. Additionally, each channel was exported separately using Steve^®^ software v.2.6 (Nanolive^®^, Tolochenaz, Switzerland) and managed with Image J Fiji v1.7 (NIH, Bethesda, MD, USA) as described elsewhere [38,39].

### 2.4. Immunofluorescence Staining

Plasmatocytes were identified using an anti-NimC1 antibody mix (diluted 1:30) containing antibodies P1a and P1b [40]. Lamellocytes were identified using the L1 anti-Atilla antibody mix (diluted 1:300) containing antibodies L1a, L1b and L1c [40]. The NimC1 and L1 antibodies were kindly provided by István Andó (Biological Research Centre, Szeged, Hungary). Crystal cells were identified using antibody HC12F6 (diluted 1:30) kindly provided by Martin Speckmann and Tina Trenczek (Justus Liebig University, Giessen, Germany). Nuclear histones within hemocyte-derived extracellular traps were detected using the global anti-histone antibody MAB3422 (Merck, Darmstadt, Germany) recognizing H1, H2A/H2B, H3 and H4 (diluted 1:1000).

Samples were washed in sterile PBS, fixed with 4% PFA for 5 min and blocked for 5 min in sterile PBS containing 2% bovine serum albumin (Merck, Darmstadt, Germany) and 0.1% Triton X-100 (Merck, Darmstadt, Germany). After incubation with the primary antibodies described above at room temperature for 1 h, binding was detected with a goat anti-mouse Alexa Fluor 555 secondary antibody (Thermo Fisher Scientific, Schwerte, Germany, diluted 1:500) at RT for 1 h. The samples were then washed in PBS and mounted in Fluoromount-G anti-fading medium (Thermo Fisher Scientific, Schwerte, Germany) for analysis by confocal microscopy on an LSM 710 instrument (Zeiss, Oberkochen, Germany) with 63× magnification and a numerical aperture of 1.2 µm. Each experiment was repeated three times (using 100 larvae per condition; obtaining *n* ≈ 5000 hemocytes). Imaging processing was performed in Image J Fiji v1.7 using merged channels plugins and restricting to minor adjustment of brightness and contrast.

### 2.5. Detection of Plasmatocyte Phagocytosis and Extracellular Traps

Phagocytosis by activated hemocytes was induced in vitro by exposure to a *P. entomophila* strain (OD 600 nm = 0.1) expressing green fluorescent protein (GFP), kindly provided by Bruno Lemaitre (École Polytechnique Fédérale de Lausanne, France) [41]. Different ODs were previously tested for the scope of this experiment; however, this led to an excess of background. To this end, OD 600 nm = 0.1 was shown to trigger the desired immune responses without interfering with the imaging background. The formation of filopodia was induced by adding 500 mg/mL LPS (Merck, Darmstadt, Germany) [42]. Bacteria or LPS were added to Grace’s insect medium containing 0.001% Hoechst and were incubated at room temperature for 1 h. The medium was then removed and hemocyte monolayers were gently washed with PBS before fixing with 4% PFA for 10 min. Plasmatocyte immunostaining was carried out as described above. Actin was stained using Texas Red-X phalloidin (Thermo Fisher Scientific, diluted 1:500) at room temperature for 90 min. For the visualization of extracellular DNA filaments, cells were stained with DAPI for 5 min. Bacteria were identified by visualizing GFP expression. Each experiment was repeated three times (using 100 larvae per condition; obtaining *n* ≈ 5000 hemocytes) and a representative image was chosen. Imaging processing was performed in Image J Fiji v1.7 using merged channels plugins and restricting to minor adjustment of brightness and contrast.

### 2.6. Scanning Electron Microscopy (SEM)

Plasmatocytes from *D. suzukii* were co-cultivated with GFP^+^ *P. entomophila* (OD 600 nm = 0.1) for 1 h on 10 mm coverslips (Thermo Fisher Scientific, Schwerte, Germany) pre-coated with 0.01% poly-L-lysine (Merck, Darmstadt, Germany) for 15 min at RT. The cells were then fixed in 2.5% glutaraldehyde (Merck, Darmstadt, Germany), post-fixed in 1% osmium tetroxide (Merck, Darmstadt, Germany) and washed in distilled water before dehydration and critical point drying with CO_2_. Finally, the cells were gold coated by sputtering and viewed on a Philips XL30 scanning electron microscope (Institute of Anatomy and Cell Biology, Justus Liebig University, Giessen, Germany). Each experiment was repeated three times (using 100 larvae per condition; obtaining *n* ≈ 5000 hemocytes) and a representative image was chosen.

## 3. Results

### 3.1. Characterization of D. suzukii Larval Hemocytes

Hemocytes isolated from *D. suzukii* L3 larvae ranged in diameter from 10 to 50 µm. The isolated cells were classified by their morphology after toluidine blue staining (Figure 1A–C) followed by staining with hemocyte-specific antibodies (Figure 1D–F). Round granular cells, presenting an average diameter of 9.69 ± 2.96 µm and ranging from 5.14 to 14.70 µm (*n* = 100) (Figure 1A), that were stained by the NimC1/P1 antibody (Figure 1D), were identified as plasmatocytes. The crystal cells were similar in size (average diameter of 9.66 ± 2.07 µm, ranging from 5.82 µm to 13.41 µm; *n* = 100) to the plasmatocytes, but were stained darkly with toluidine blue due to the presence of crystals in the cytoplasm (Figure 1B). They were also stained by the C1-specific antibody, which reacts with prophenoloxidase 2 (PPO2) in *Drosophila* spp. [32] (Figure 1E). The lamellocytes ranged in morphology from oval to elongated forms (average diameter of 28.24 ± 9.66 µm, ranging from 17.89 µm to 47.89 µm; *n* = 100) with a dark nucleus (Figure 1C), but they could be identified by staining with the L1 Atilla-specific antibody (Figure 1F). The most abundant cells (Figure 1G) were plasmatocytes (89.9%), followed by crystal cells (7.5%) and lamellocytes (2.6%).

### 3.2. Viability of Freshly Isolated D. suzukii Larval Plasmatocytes

The viability of plasmatocytes from *D. suzukii* L3 larvae was assessed by 3D holotomography [38]. Freshly isolated cells were incubated in an imaging medium at room temperature until more than 90% of the cells were dead. At the beginning of the incubation period, plasmatocytes were generally rounded (Figure 2, RI), with a central nucleus stained with Hoechst (Figure 2, DNA). Visible granules surrounding the nucleus registered higher RI values than the rest of the cell contents (Figure 2, zoomed images). After 2 h of isolation, circa 50% of the cells remained viable (Figure 2, absence of SYTOX Green staining) but 90% of the cells were stained with SYTOX Green after 4 h (Figure 2, survival rate), indicating cell death, even though the cells maintained their shapes. Accordingly, all subsequent experiments were limited to 2 h post-isolation to ensure that most of the freshly isolated plasmatocytes were viable.

### 3.3. Response of D. suzukii L3 Plasmatocytes to P. entomophila

The exposure of *D. suzukii* plasmatocytes to a GFP^+^ *P. entomophila* strain triggered the formation of extracellular trap structures with a range of phenotypes (Figure 3), as described for other species elsewhere [42,43,44]. Electron microscopy confirmed that *D. suzukii* plasmatocytes react against *P. entomophila*, not only by casting extracellular traps, but also by forming filopodia (Figure 3).

The plasmatocytes were able to cast short spread extracellular traps (*spr*ETs) that captured the cocultured bacteria (Figure 3A, green arrows). Some filigree filaments (Figure 3A,B, blue arrows) were also attached to bacterial cells, but not to hemocytes. These were probably extracellular DNA filaments derived from *spr*ETs that were damaged by the evasion attempts of bacteria or by experimental handling. We also observed the presence of so-called aggregated extracellular traps (*agg*ETs) in response to GFP^+^ *P. entomophila* (Figure 3B,C). These formed large meshes of extracellular fibers containing many immune cells releasing individual extracellular traps (Figure 3C, green arrows) and were able to trap several bacteria at once (Figure 3B, white arrows).

To confirm that *D. suzukii* plasmatocytes cast extracellular traps, we stained the typical components of such structures in immunofluorescence assays. After challenging plasmatocytes with *P. entomophila* or 500 mg/mL LPS, the formation of extracellular traps was confirmed by the co-localization of extracellular DNA and histones (Figure 4). Interestingly, LPS induced short *spr*ETs (Figure 4, blue arrow) and diffuse extracellular traps (*diff*ETs), the latter characterized by nuclear expansion and thus cell expansion with histone redistribution (Figure 4, red arrows). We observed the same response to three different concentrations of LPS (100, 250 and 500 µg/mL). In contrast, *P. entomophila* induced *spr*ETs (Figure 4, blue arrows) and *agg*ETs (Figure 4, green arrows), both of which were shown to entrap the GFP^+^ bacteria (Figure 4, white arrows), confirming the SEM data.

The percentage of plasmatocytes that produced extracellular traps was calculated after coculture with *P. entomophila* or stimulation with LPS (Figure 5). Approximately 9% of plasmatocytes produced extracellular traps in response to *P. entomophila*, whereas only 3.6% produced extracellular traps in response to stimulation with LPS, although the difference was not statistically significant (Figure 5A). Additionally, different ETs were displayed when cells were stimulated after each condition (Figure 5B). In response to *P. entomophila*, 41% of the displayed ETs represented *spr*ETs, whereas 59% displayed *agg*ETs (Figure 5B). In addition, 77% of the LPS-stimulated plasmatocytes displayed *diff*ETs whereas 23% displayed *spr*ETs.

To confirm that the observed immunoreactive behavior was cast by *D. suzukii* plasmatocytes, plasmatocyte-specific anti-NimC1 antibody staining was used. Positive stained cells were the ones releasing extracellular traps (Figure 6, white asterisk). The *D. suzukii* plasmatocytes also engulfed *P. entomophila* by phagocytosis (white arrows). One hour after the challenge, 9.86% of the plasmatocytes were shown to engage in the phagocytosis of *P. entomophila* (Figure 7, orange arrow).

In addition to phagocytosis and extracellular traps, we also observed the formation of filopodia as a third effector mechanism against pathogenic bacteria (Figure 3D, red arrow), followed by the adhesion of bacteria to these structures (Figure 8). Texas Red-X phalloidin staining (Figure 8, phalloidin) confirmed the actin-dependent formation filopodia in response to the bacteria and LPS, resulting in the presentation of rounded to slightly elongated plasmatocytes.

## 4. Discussion

In insects and other invertebrates, hemocytes are the first line of defense, eliminating pathogens upon first encounter in vivo and therefore fulfilling a similar role to professional phagocytes (PMNs, monocytes and macrophages) in mammals. We characterized the hemocyte population of the invasive pest insect *D. suzukii* to add to the understanding of diverse effector mechanisms as part of the early innate immune response, which has not been reported for other drosophilid species thus far. We observed phagocytosis and the formation of filipodia by *D. suzukii* plasmatocytes exposed to *P. entomophila* or LPS, but also the formation of extracellular traps (ETosis), highlighting the importance of this widespread cellular immune defense mechanism in eukaryotes [23,45]. ETosis has been reported in other insects [33,46], and also in other invertebrates, such as oysters [47,48], mussels [46] and slugs [29], and the mechanism appears highly conserved. The traps are composed of extracellular DNA decorated with nuclear histones (chromatin), combined with lactoferrin, pentraxin, myeloperoxidase, elastase, gelatinase and cathelicidin, among other antimicrobial molecules [22,23,24,26,40]. Interestingly, we observed the presence of multiple histones (H1, H2A/H2B, H3 and H4), whereas one recent study in the cockroach *Periplaneta americana* only identified H1/DNA complexes in extracellular traps [42], with only one study in slugs supporting our findings [29]. However, our results align with the mechanism of ETosis in mammals, where histones are one of the main components of extracellular traps both in vitro and in vivo [46]. The formation of extracellular traps by drosophilid hemocytes has not been reported before, and the absence of nucleic acid clots in the hemolymph of *D. melanogaster* was reported following a challenge with *Escherichia coli* JM109 [49]. The nature of the pathogenic stimulus may determine the mode of cellular defense. Here, we used *P. entomophila*, a bacterial pathogen that has already been shown to cause systemic infections that induce a range of immune responses in drosophilids [50,51]. The bacteria can be highly pathogenic when flies receive inoculum sizes sufficient to disrupt the gut epithelium and enter the hemolymph, which brings them into contact with hemocytes [50,51].

We characterized the morphology of *D. suzukii* hemocytes in detail and observed similar characteristics to the closely related fly *D. melanogaster* [13]. We were able to distinguish between lamellocytes, crystal cells and plasmatocytes using both morphological criteria and specific immunostaining, which allowed us to demonstrate the unique defense mechanisms of the plasmatocyte population. The revelation that such cells can form filopodia and extracellular traps in response to *P. entomophila* supports findings in other insects [42]. The initial detection of pathogenic bacteria is facilitated by the recognition of pathogen-associated molecular patterns (PAMPs) such as LPS. Our results suggest that PAMPs alone, which bind pathogen recognition receptors (PRRs) on the surface of plasmatocytes, are sufficient for inducing the formation of filopodia and the release of extracellular traps. Similar (dose-dependent) effects have been described in the cockroach *P. americana* [42] when hemocytes are stimulated with delipidated LPS. Naturally, hemocyte PRRs sense and respond differently to LPS or infections with live bacteria [52], highlighting the importance of adjusting the bacterial titer in order to study humoral and cellular immune responses. However, future studies should address the effect of delipidated LPS and/or other bacterial strains on the *D. suzukii* cellular response. The potential role of hemocyte “trained immunity” after a primary infection with *P. entomophila* is also an interesting topic because this form of innate immune memory has been well described in mammals and, despite the long-believed lack of “immune priming” capacities in invertebrates, more recent studies indicate its presence also in insects [53,54,55,56].

Interestingly, *D. suzukii* plasmatocytes-derived ETosis revealed up to three different types of extracellular traps, namely *spr*ETs, *agg*ETs and *diff*ETs, as previously described for other hosts and cells [29,43]. Exposure to bacteria or LPS induced the release of *spr*ETs, whereas *diff*ETs were only observed following a challenge with LPS. In addition, *agg*ETs were cast after *P. entompohila* stimulus. Neutrophils are known to discriminate between LPS and bacterial infection, releasing NETs that differ in structure and activity [57]. In mammals, it is likely that *spr*NETs and *diff*NETs are preliminary structures, whereas *agg*NETs are later and more mature forms [43]. Further experiments with longer exposure times would be necessary in determining whether a similar temporal profile exists in insects with the presence of even larger or a higher number of *agg*ETs. It would also be interesting to investigate whether insect *agg*ETs inhibit inflammation by degrading cytokines and chemokines as they do in mammals [44].

*D. suzukii* robust cellular immune responses might have facilitated its rapid worldwide spread by allowing it to overcome pathogen infections in newly colonized environments [55,56,57]. Such immunological diversity can help invasive species deal with unfamiliar pathogens [10,11,12,13,14,15,16,17,18,19,20,21,22,23,24,25,26,27,28,29,30,31,32,33,34,35,36,37,38,39,40,41,42,43,44,45,46,47,49,50,51,52,53,54,55,56,57,58], as recently shown for the invasive harlequin ladybird *Harmonia axyridis* [48,55,56,57,58,59,60,61]. We therefore propose that extracellular traps are a key component of the *D. suzukii* cellular innate immune response against pathogenic bacteria. The similarity between vertebrate and invertebrate cellular immunity highlights the evolutionary conservation of this ancient mechanism.

## Figures and Tables

**Figure 1 cells-10-03320-f001:**
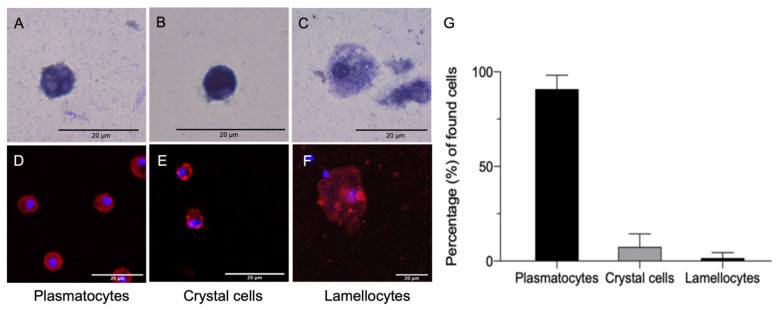
Characterization of hemocytes from *Drosophila suzukii* L3 larvae. Cells isolated from the hemolymph of *D. suzukii* L3 larvae were fixed with 4% paraformaldehyde for 5 min at room temperature and stained with 1% toluidine blue (**A**–**C**) followed by immunofluorescence staining (**D**–**F**). Plasmatocytes (**A**), identified by staining with the anti-NimC1 antibody (**D**), were rounded cells with central nuclei. Crystal cells (**B**), identified by staining with antibody HC12F6 (**E**), were rounded cells containing crystals that were stained densely with toluidine blue. Lamellocytes (**C**), identified by staining with the L1 anti-Atilla antibody (**F**), were oval or elongated cells. Nuclei were counterstained with DAPI (**D**–**F**, blue). Cell population analysis (**G**) revealed plasmatocytes to be the most abundant cells (89.9%), followed by crystal cells (7.5%) and lamellocytes (2.6%). Scale bar = 20 μm.

**Figure 2 cells-10-03320-f002:**
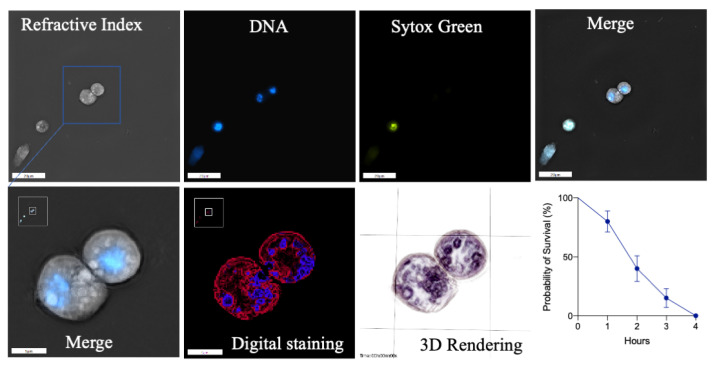
Plasmatocyte viability assay and characterization by 3D holotomography. Plasmatocytes were mostly rounded cells (RI) with a central nucleus (DNA) and perinuclear granules (zoomed images on bottom row). SYTOX Green staining, which is specific for dead cells, showed that circa half of the cells remained viable for at least 2 h but that 90% were dead after 4 h. Scale bar = 5 μm.

**Figure 3 cells-10-03320-f003:**
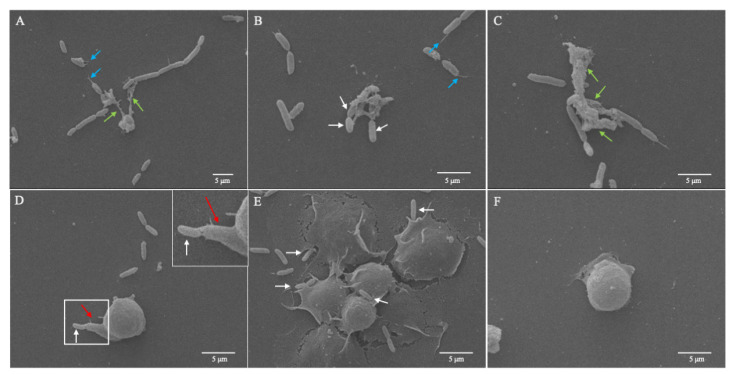
SEM images of extracellular traps formed by *Drosophila suzukii* plasmatocytes in response to *Pseudomonas entomophila.* The isolated plasmatocytes were incubated with *P. entomophila* (**A**–**E**) or PBS (control, **F**) at room temperature for 1 h. The plasmatocytes cast short spread extracellular traps (*spr*ETs) to entrap *P. entomophila* (**A**, green arrows). Filigree filaments (**A**,**B**, blue arrows) from disrupted extracellular DNA filaments also attached to *P. entomophila*. So-called aggregated extracellular traps (*agg*ETs) were observed when cells were challenged with *P. entomophila* (**B**,**C**), resulting in massive clusters of mesh fibers containing many immune cells (**C**, green arrows) and bacteria (**B**, white arrows). The formation of filopodia (**D**) was observed after further incubation, and *P. entomophila* was found stuck to these structures (**D**,**E**, white arrows). Scale bar = 5 μm.

**Figure 4 cells-10-03320-f004:**
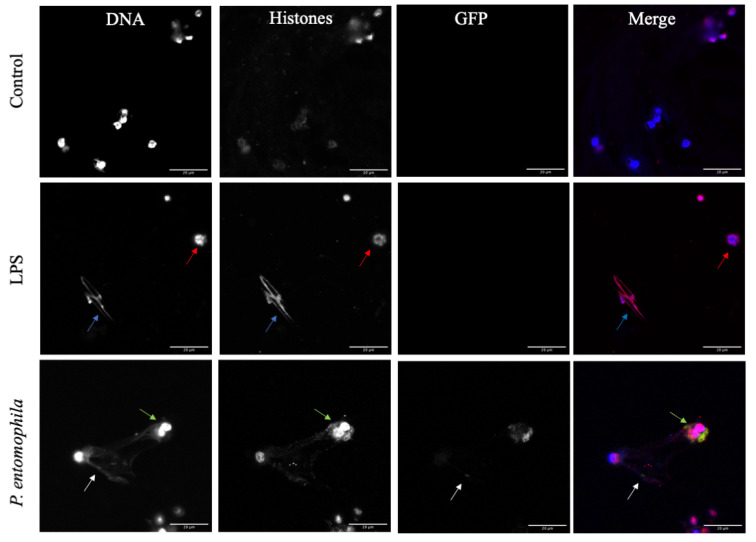
Characteristics of extracellular traps formed by *Drosophila suzukii* plasmatocytes in response to *Pseudomonas entomophila* and lipopolysaccharides (LPS). Isolated plasmatocytes were incubated with GFP^+^ *P. entomophila*, LPS or PBS as a control for 1 h before fixing with 4% PFA for 5 min at room temperature and staining with Hoechst (blue). Histones (H1, H2A, H2B, H3 and H4) were then detected with the monoclonal antibody MAB3422 followed by staining with the goat anti-mouse IgG Alexa Fluor 555 (red) and nuclear counterstaining with DAPI (blue). One hour after the challenge, two different phenotypes were observed. LPS induced spread extracellular traps (*spr*ETs, blue arrows), as well as diffuse extracellular traps (*diff*ETs, red arrows), whereas *P. entomophila* (white arrows) induced *spr*ETs (blue arrows) and aggregated extracellular traps (*agg*ETs, green arrows). Scale bar = 20 μm.

**Figure 5 cells-10-03320-f005:**
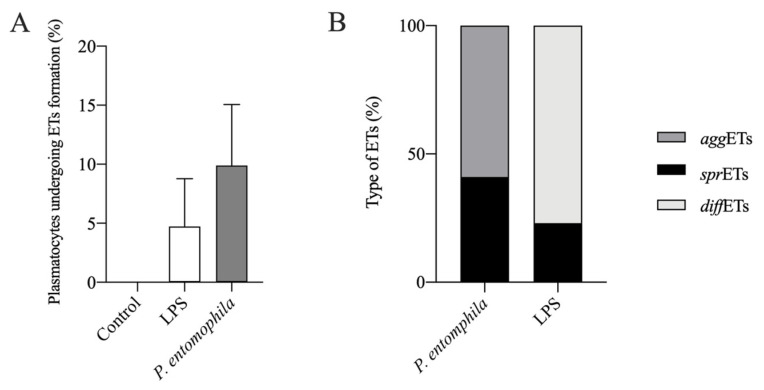
Quantification of extracellular traps induced by *P. entomophila* or LPS. (**A**) We compared the proportion of plasmatocytes that cast extracellular traps in response to GFP^+^ *P. entomophila* and LPS. *P. entomophila* induced higher ET formation than LPS when compared to control (no ET formation). (**B**) The three different ETs phenotypes were quantified and LPS induced *spr*ETs, as well as *diff*ETs (in greater percentage), whereas *P. entomophila* induced *spr*ETs and *agg*ETs, the latter in higher proportion.

**Figure 6 cells-10-03320-f006:**
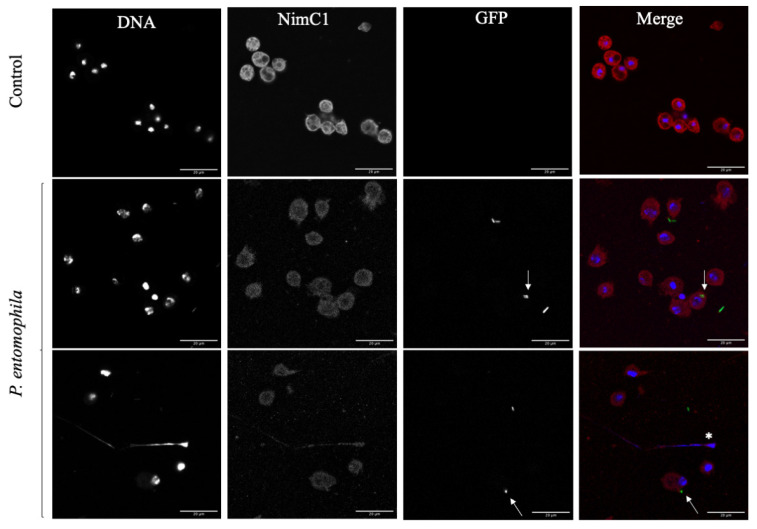
*Drosophila suzukii* plasmatocytes cast extracellular traps and engulf *Pseudomonas entomophila* by phagocytosis. Plasmatocytes were challenged with GFP^+^
*P. entomophila* and stained with the plasmatocyte-specific antibody anti-NimC1 to confirm their identify. We observed the formation of extracellular traps (white asterisk, lower row) but also the phagocytosis of *P. entomophila* (white arrows, middle and lower rows), indicating that plasmatocytes respond to the challenge by deploying multiple defense mechanisms. Scale bar = 20 μm.

**Figure 7 cells-10-03320-f007:**
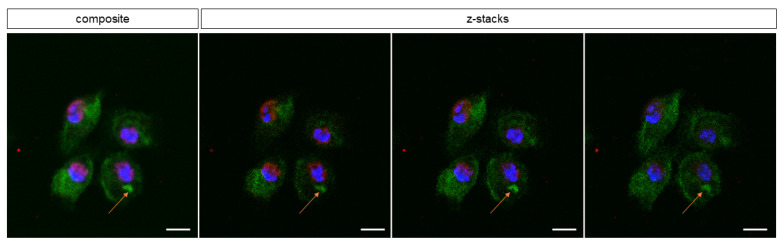
*Drosophila suzukii* plasmatocytes phagocyte *Pseudomonas entomophila*. After plasmatocytes were challenged with GFP^+^
*P. entomophila*, bacteria were observed intracellularly near to the nucleus of plasmatocytes (orange arrows), showing that plasmatocytes use multiple defense mechanisms, including phagocytosis. Nucleus was stained with DAPI (blue), histones in red and bacteria in green. Auto-fluorescence of the cell is observed. Scale bar = 10 μm.

**Figure 8 cells-10-03320-f008:**
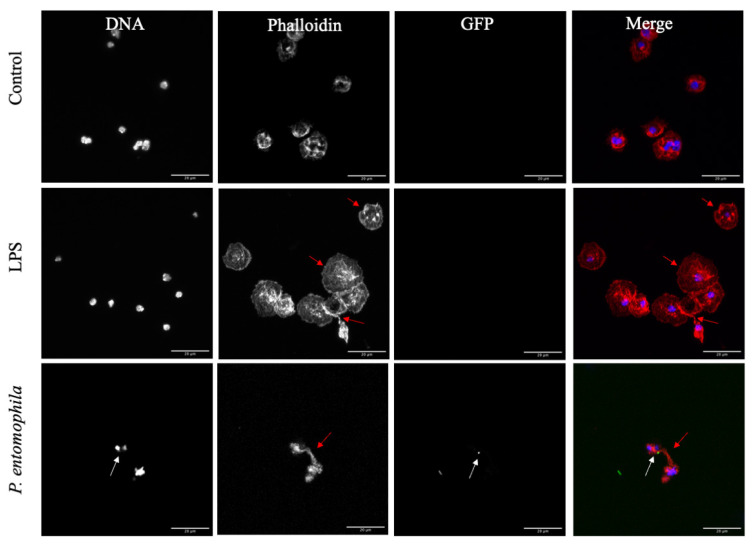
Formation of filopodia by *D. suzukii* plasmatocytes. Isolated plasmatocytes were incubated with GFP^+^
*P. entomophila*, 500 µg/mL LPS or PBS as a control for 1 h before fixing in 4% paraformaldehyde for 5 min at room temperature and staining for DNA with Hoechst (blue) and for actin with Texas Red-X phalloidin (Phalloidin, red). Filopodia (red arrows) formed in response to both LPS and *P. entomophila* (white arrows), resulting in the presentation of rounded to slightly elongated cells (arrow). Scale bar = 20 μm.

## Data Availability

Not applicable.

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
