# Peer review of "The Cellular Innate Immune Response of the Invasive Pest Insect Drosophila suzukii against Pseudomonas entomophila Involves the Release of Extracellular Traps"

_cells, 2021, doi:10.3390/cells10123320_

Round 1
Reviewer 1 Report
In this paper the authors describe the morphology of hemocytes in Drosophila suzukii and, for the first, identify extracellular nets which have not been previously described in this species. The paper is well written and to the point and provides useful information for identifying hemocytes in D. suzukii, a common agricultural pest. This study will set the stage for important immunological studies as well as inform the use of biological control agents against this pest. While the authors provide some interesting data as well as newly identified kinds of cellular defense in the insect, more experimentation and clarification would strengthen the paper. I recommend to reconsider after major revision.
Comments:
Methods:
- how many replicates were performed for all the representative images?
- “Finally the cells were sputtered with gold particles” … is sputtered the right word?
Results:
- “ranged in diameter” … if the point is to characterize hemocytes it would be better to measure cell diameter across X cells and get an average diameter per cell type
- Figure 3: can you show a percentage of cells staining with cytox over time? Graphical representation would be nice to see. As well as representative images per time point. Please show quantifications.
- In results it says “more than half” were viable whereas in the figure 2 legend it says “half of cell were viable” at 2 hours. Please quantify and demonstrate which is it… if 2 hours was used as a time point for further analyses then is 50% viability sufficient?
- Figure 3: what is the bacteria used in A and B if not p entomophila?
- 7. Figure 3. Last line of legend: “The formation of filopodia (D, red arrow) was observed after further incubation, and P. entomophila was found stuck to these structures (E and F, white arrows). Scale bar = 5 μm.”… there are no arrows in f. also isn’t it the control?
- 8. Figure 4. “ LPS induced spread [of] extracellular traps”
- 9. Figure 5. How did you quantify exracelleular traps? Also were all types of ETs included? Agg? Spr? diff? was there a different proportion of cells which made these? Do you see heterogeneity in the plasmatocytes which may indicate different functions? Show representative photo demonstrating what was measured and how if applicable. Is it time dependent? Do you see more cells engaging in ET formation over time?
- 10. Figure 6. This confirms that plasmatocytes create nets but it doesn’t say anything about what ccs or lamellocytes are doing. Confirmation bias. Negative stainings for the other cells with lack of ETs would be appropriate.
- 11. Figure 6. More labels are necessary. What is the difference between the middle row and last row? How can you tell the entomophila is engulfed? Show a z stack or 3d representation?
- Figure 7. Where are the labels or arrows for the filopodia? Which bacteria in the figure legend?
Discussion:
- “higher eukaryotes” is a dated concept which is human centric. Redescribe in terms of non-biased phylogeny ordering.
- “Interestingly, we observed the presence of multiple histones (H1, H2A/H2B, H3 and H4) ” when and how did you observe this?
- ”innate immune memory has been described in mammals and could therefore be also present in insects [52]”… there are quite some studies now already supporting this in D. melanogaster.
- “It is likely that sprNETs and diffNETs are preliminary structures, whereas aggNETs are later and more mature forms ”… did you name these forms or have they been previously described in other species?
Reviewer 2 Report
The authors provide evidence to demonstrate that Drosophila suzukii plasmatocytes deploy Etosis in vitro in response to a bacterial suggesting that this effector mechanism could be relevant in defense of D. suzukii against potential threats.
I consider that the authors have done a remarkable job demonstrating that D. suzukii plasmatocytes cells respond to bacterial challenges using the mechanism of etosis. This mechanism has been well characterized in vertebrates, but it has been described only in a few species of invertebrates. Furthermore, carrying out this type of work requires dedication and technical skill due to the challenges posed by isolating hemocytes from the hemolymph and manipulating cells of such small size.
Furthermore, the article has a well-defined structure that makes it clear and pleasant to read. However, I have a few comments:
1) In the introduction, the authors describe the importance of D. suzukii as a pest insect, which is necessary to offer context to the readers. However, the fragment:
Biological pest control measures based on natural predators such as parasitoid
wasps (e.g., Leptopilina heterotoma and Asobara japonica) have not been successful because
the flies benefit from natural resistance to parasitoid larvae [8–10]. This innate resistance
appears to reflect the high hemocyte count of infected individuals and efficient
hemocyte recruitment to infection sites [9,10]
It is not relevant for the manuscript since the authors do not carry out any test in which it is demonstrated that Etosis is deployed against the eggs or the particles injected to the host during oviposition of parasitoid wasps. Likewise, biological control is not a relevant aspect in the paper. Therefore, this fragment is not connected with the rest of the manuscript. Furthermore, I consider that instead of providing the necessary context to address the central objective of the manuscript, it could convey to the readers a wrong expectation about the content of the article.
2) In the introduction, the authors state:
The systemic infection of D. suzukii by Pseudomonas entomophila leads to the accumulation
of bacteria within the hemolymph, resulting in encounters between these bacteria
and circulating hemocytes
However, it is unclear whether this observation is acquired by the authors or by others, making it necessary to clarify this point.
3)In the discussion, the authors begin with the following statement:
In vertebrates, the formation of NETs is an ancient and well-conserved innate immune
defense mechanism against pathogens [45]. In insects and other invertebrates, hemocytes
are the first line of defense, eliminating pathogens upon first encounter in vivo
and therefore fulfilling a similar role to professional phagocytes (PMNs, monocytes and
macrophages) in mammals
the introduction provided the general context to the reader. Therefore, I suggest eliminating and limiting the discussion to the actual results of the paper.
The same happens with the statement:
There are three different hemocyte lineages that arise from the head mesoderm during
early drosophilid embryogenesis [14].
Which is not linked to the results
4) In the discussion the authors make the following statement:
We characterized the hemocyte population of the invasive pest insect D. suzukii to understand the diverse effector mechanisms as part of the early innate immune response.
This statement is not correctly supported by the data presented in the article since the authors only established the percentages of the three types of hemocytes, and it is clear that in the article, the only mechanism of immunity evaluated is etosis. Although phagocytosis was detected, it appeared more as an incidental observation since there was not a specific set of experiments designed to assess this effector mechanism. However, in insects, the innate immune response presents various effector mechanisms that were not evaluated, such as the production of ROS that in Drosophila occurs within the first hour after a challenge with bacteria.
5) In the discussion, the authors state
Plasmatocytes can detect foreign bodies such as parasitoid wasp eggs laid in the larval hemocoel and encapsulate them [44], ultimately killing the parasitoid by hypoxia or the local production of ROS, quinones, or semiquinone
However, this statement is not directly linked to the results of the work presented in the paper. Hence I suggest removing it.
6) The authors present the following statement:
Our results suggest that the robust cellular response of the invasive pest species D.
suzukii may have facilitated its rapid worldwide spread by allowing it to overcome pathogens
and parasites in newly colonized environments.
I cannot entirely agree with this statement since the authors do not carry out any in vivo experiment in which it is proven that the effector mechanism of etosis is a relevant protective response for D. suzukii once it faces a challenge with bacteria. On the other hand, talking about pathogens, in general, is risky since only a bacterium and a bacterial molecular pattern were tested in this study. I suggest rewriting this phrase to fit the scope of the study

Reviewer 3 Report
Here, Tessa Carrau et al worked with primary cultures of Drosophila suzukii hemocytes and evaluated whether the immune defense of these cell types would involve the production of extracellular traps. Overall, the manuscript is concise and well-written. On the other side, I found that the description of the methodology could be improved for more detail. Finally, while this is an interesting field of research, I encourage the imaging and some other issues to be further developed before the manuscript is ready for publication. Ideally, they would also provide more information on the biological significance of extracellular traps in this model.
Major issues
Methodology
I was not sure I followed the protocol from hemocyte collection. Was something done between “Third-instar larvae (L3) were washed up to 10 times in distilled water to remove debris” and “Hemocytes were isolated directly in Nunc Lab-Tek II chamber slides (…)”. I did not understand how hemocytes were taken from inside the larvae and put in cell media.
Fig 4, 6, 7:
Overall, the authors work with a very small number of cells and cellular events in the field of view. At the current form, it is difficult to evaluate whether the images being shown match the quantifications described. For example, in Fig 7, it seems that only 2 cells are present in the Pseudomonas group. What is the reason for that? Could they increase the number of cells in the field of view by increasing the number of larvae used per condition tested?
The number of GFP bacteria is relatively small. How were these conditions optimized? Interestingly, I was only able to find information concerning the protocols of P. entomophila co-culture for the SEM experiments. Similarly, how was the concentration of the LPS assay optimized?
Magnification from Fig 4, 6, 7 is relatively low. While this can be useful when counting cells and cellular events, it is not very informative when analyzing the image. Could the authors provide higher-magnification images?
Fig 6
“To confirm that the observed immunoreactive behavior was restricted to D. suzukii plasmatocytes, we stained the cocultures with the plasmatocyte-specific anti-NimC1 antibody, clearly showing that the stained cells released the extracellular traps (Fig. 6, white asterisk).” I need immunolabeling images from other cell types in order to interpret whether only plasmatocytes are immunoreactive.
Minor issues
More information is needed concerning sample size in all the experiments.
Did the authors try to use DNAse treatment to validate their results?
What is the importance of extracellular traps during the immune response of D. Suzukii.
Round 2
Reviewer 1 Report
the authors have adequately addressed the comments and the paper has been strengthened. it will be a useful resource to the scientific community particularly regarding innate immunity and understanding agricultural pests.
Reviewer 3 Report
The authors addressed my remarks and provided improvements in the text of the manuscript and its results. In the current form, I recommend this manuscript for publication.